# Deep Q-Learning with Bit-Swapping-Based Linear Feedback Shift Register fostered Built-In Self-Test and Built-In Self-Repair for SRAM

**DOI:** 10.3390/mi13060971

**Published:** 2022-06-19

**Authors:** Mohammed Altaf Ahmed, Suleman Alnatheer

**Affiliations:** Department of Computer Engineering, College of Computer Engineering & Sciences, Prince Sattam Bin Abdulaziz University, Al-Kharj 11942, Saudi Arabia; s.alnatheer@psau.edu.sa

**Keywords:** built-in self-test, Built-In Self-Repair, Built-In Redundancy Analysis, deep Q-learning, fault injection, system-on-chip, CMOS, Static Random Access Memory

## Abstract

Including redundancy is popular and widely used in a fault-tolerant method for memories. Effective fault-tolerant methods are a demand of today’s large-size memories. Recently, system-on-chips (SOCs) have been developed in nanotechnology, with most of the chip area occupied by memories. Generally, memories in SOCs contain various sizes with poor accessibility. Thus, it is not easy to repair these memories with the conventional external equipment test method. For this reason, memory designers commonly use the redundancy method for replacing rows–columns with spare ones mainly to improve the yield of the memories. In this manuscript, the Deep Q-learning (DQL) with Bit-Swapping-based linear feedback shift register (BSLFSR) for Fault Detection (DQL-BSLFSR-FD) is proposed for Static Random Access Memory (SRAM). The proposed Deep Q-learning-based memory built-in self-test (MBIST) is used to check the memory array unit for faults. The faults are inserted into the memory using the Deep Q-learning fault injection process. The test patterns and faults injection are controlled during testing using different test cases. Subsequently, fault memory is repaired after inserting faults in the memory cell using the Bit-Swapping-based linear feedback shift register (BSLFSR) based Built-In Self-Repair (BISR) model. The BSLFSR model performs redundancy analysis that detects faulty cells, utilizing spare rows and columns instead of defective cells. The design and implementation of the proposed BIST and Built-In Self-Repair methods are developed on FPGA, and Verilog’s simulation is conducted. Therefore, the proposed DQL-BSLFSR-FD model simulation has attained 23.5%, 29.5% lower maximum operating frequency (minimum clock period), and 34.9%, 26.7% lower total power consumption than the existing approaches.

## 1. Introduction

The primary purpose of memories in systems-on-chips is “to store huge amounts of data” [1]. Here, the memories occupy more space in SOC design, developed in CMOS technology. It contains a smaller feature size [2], which specifies that memories are essential for yield impact [3,4]. The memory repair principle includes either row or column repair, or both [5]. Traditionally, memory repair is executed in two stages. The first stage is examined the failure identified by the memory built-in self-test controller through the test of repairing memory. The second stage determines the repair signature to repair memory [6,7]. Each repairable memory has repair registers, which hold the repair signature. Additionally, the BIRA method calculates the repair signature according to memory failure data and executes the memory redundancy model. It defines whether the memory is repaired in the production testing platforms. Therefore, the repair signature is stored in the Built-In Redundancy Analysis registers for processing MBIST Controllers [8,9,10,11].

The repair signature is sent to the scan chain of the repair registry for the chip programming placed at the chip design level [12]. Here, the fusebox’s read and write test access port (TAP) is controlled, and the special repair logs record the scan chains that connect the memories to the fuse. Moreover, the repair data is scanned from the scan chains, compressed, and burned while flying in the eFuse line using higher voltage nuts. On-chip reset, restoration data from eFuse is loaded spontaneously and debugged into repair logs attached to the memory [13]. As a result, all memories are fixed by redundancies. Finally, MBIST runs on repaired memory, which verifies the correctness of memory.

Usually, memories do not contain logic gates and flip-flops. So, various fault models and test algorithms are needed to test memories. If there are any faults in memory, it affects the overall output of SOC. Due to this reason, the spare rows and columns are included in memory. Where erroneous cells are needed to run, spare rows/columns are utilized in the place of damaged cells using BISR logic. The repair logic contains line/column repair or both. Then, the memory built-in self-testing and repair tool tests memory effectively to detect possible faults among normal memory cells. Thus, the below fault models are adequate to test the memory, such as Stuck-at-Fault, Transition Fault, Coupling Fault, Neighborhood Pattern Sensitive Fault (NBSF), and Address Decoder Faults [14,15,16,17,18]. Furthermore, error detection methods provide a way to detect faulty sites, and existing redundancy analysis (RA) methods offer repair solutions for these defective sites [19]. Moreover, repair solutions include spare rows and columns for misplacement of memory.

Furthermore, the rows and columns pivots and repair requests are serviced based on the precedence list created through actions to reduce complexity and tracing time based on execution and finding row and column pivots. However, it does not reduce the power consumption, area, and time while implementing FPGA [20]. A simple BIST and BISR method is presented in the research [21,22]. It uses an MBIST controller and BISR algorithm for testing memory and is implemented on ASIC, and the March algorithm is used in the research to test the MUT. In 2021, Fragkos presented an artificial intelligence-based approach using a large test-bed architecture [23] to deal this the fault detection problem. Although, some optimization techniques still are required to minimize these issues for optimal memory built-in self-test of SRAM. Thus, the Deep Q-learning (DQL) with Bit-Swapping-based linear feedback shift register (BSLFSR) model is proposed in this manuscript for SRAM testing using MBIST and BISR methods.

The main contributions of this manuscript are described as follows:

In this manuscript, Deep Q-learning (DQL) with Bit-Swapping-based linear feedback shift register (BSLFSR) for Fault Detection (DQL-BSLFSR-FD) is proposed for SRAM.

Then, the proposed hybrid technique is the combined execution of both the Deep reinforcement learning (DQL) [24] and Bit-Swapping-based linear feedback shift register (BSLFSR) [25]; hence it is named the DQL- BSLFSR technique.

In this manuscript, the Deep Q-learning (DQL)-based BIST is used to check the memory array circuit and injects faults into memory.

Additionally, BSLFSR-based BISR is used for repairing faulty memory cells based on BIRA. After inserting faults in the memory cell, fault memory is fixed with BSLFSR for the Built-In Self-Repair scheme.

Subsequently, the simulation and synthesis outcomes are attained by utilizing Mentor Graphics and Xilinx ISE 14.5 design suite.

Furthermore, the design and implementation of FPGA architecture by memory built-in self-test and self-repair hardware structure for Static Random Access Memory are tested using Verilog.

Finally, the performance measures are analyzed, such as power, area, and operating frequency of BIST and BISR of the proposed DQL-BSLFSR-FD method.

The simulation outcomes of the DQL-BSLFSR-FD method are compared with existing methods, such as Extreme Learning Machine-based Autoencoder for fault diagnosis (ELM-AE-FD) [26], and built-in self-test (BIST) model-based post package inspections (PPI) for fault diagnosis (BIST-PPI-FD) [27], respectively.

Therefore, the FPGA performance of the DQL-BSLFSR-FD method is compared with the existing approaches. The existing techniques, such as the first one, the memory BIST with optimized BISR for fault detection FPGA (MBIST-OBISR-FD) [28], the second one, the SRAM-based Physically Unclonable Function (PUF) with Hot Carrier Injection (HCI) for fault detection FPGA (PUF-HCI-FD) [29], and third, Essential Spare Pivoting (ESP) based Local Repair Most (LRM) FPGA (ESP-LRM-FD) [30], respectively, are compared with the proposed method.

The remaining manuscript is structured as follows: Section 2 delineates the related works, Section 3 elaborates the proposed methodology, Section 4 demonstrates the results and discussion, and Section 5 concludes the manuscript.

## 2. Related Work

Numerous MBIST strategies were used in the previous literature. Some of the most recent research works are reviewed here.

In 2020, Kalpana et al. [26] presented an Extreme Learning Machine-based Autoencoder for parametric fault diagnosis (ELM-AE-FD) in the analog circuit. Additionally, Single and multiple parametric fault analyses were considered for simulation before the test model. Here, the features were collected to create a fault dictionary that involves different faulty and fault-free configuration values. Subsequently, the transfer function model was simulated using Monte-Carlo analysis, and the model’s performance was compared with self-adaptive evolutionary ELM approaches. Therefore, the ELM-AE-FD method achieved higher diagnosis accuracy, but the power consumption was high.

In 2021, Gopalan and Pothiraj [28] presented memory BIST with optimized BISR for fault detection in SOCs design. The BIST model checks the memory array circuit, and the optimized BISR repairs the faulty memory cells. Additionally, faults were inserted into the memory using saboteurs fault injection and mutants fault injection methods by optimized test pattern logic. After that, the fault memory was repaired by the counting threshold algorithm. Therefore, the performance of the MBIST-OBISR-FD model improved in this method, but the device utilization was very high.

In 2020, Liu [29] presented SRAM-based Physically Unclonable Function (PUF) with Hot Carrier Injection (HCI) for fault detection with high stability and low power. The PUF model utilizes hybrid operations based on CMOS-SRAM mode and Enhancement-Enhancement (EE) Static Random Access Memory mode. The PUF model was fabricated in standard CMOS at 130 nm, which achieved a high bit error rate (BER). Additionally, the accelerated aging test performs a long-term reliability operation, which enhances the model’s performance. However, fault diagnosis was not reasonable using the PUF-HCI-FD model.

In 2020, Ryabtsev and Volobuev [30] presented the BISR framework to restore RAM operability on SOCs. Hence, the given model was processed when various failures were due to backup and main memory reconfiguration. Thus, the technical solution lessens the product’s weight than most redundant devices because not every memory was allocated, but only vital components were very susceptible to failures. Therefore, the functional state of the digital SOC’s memory was automatically restored without the contribution of the framework. The BISR and BIST RAM were used for industrial and special purposes that achieved high performance in failure detection, but the area of the model was comparatively high.

In 2020, Zhou et al. [31] presented the design of the Spin-Transfer Torque-based magneto-resistive random access memory (STT-MRAM) model for the BIST process. In this approach, the tunneling magneto-resistance (TMR) was checked, and a real-time built-in self-test was triggered during the sensing operation for lasting damage in the magnetic tunnel junction (MTJ) stack. Further, the presented design was involved in magneto-resistive random access memory array execution for calculating the feasibility of the test scheme. Therefore, the designed model achieved lower power consumption and higher reliability, but the operating time of the model was high.

In 2021, Park et al. [27] presented the BIST model to process post-package inspections (PPI) for attaining fault-free Dynamic Random Access Memory (DRAM). Here, the compact and higher test coverage structures for in-DRAM-BIST were considered for resolving the area issues when applied to commodity Dynamic Random Access Memory. Subsequently, the built-in self-test model safeguards the test coverage for a short period, diminishing the test time and improving the area overhead, but still, the power utilization of the model was high.

In 2019, Pundir [25] presented improved modified memory Built-In Self-Repair (MMBISR) for Static Random Access Memory utilizing hybrid redundancy analysis (HRA). The augmented version of ESP and LRM provides a better solution for an optimized set of rows and columns appropriate to the repair process. Additionally, the fault dictionary was updated or fixed concurrently in the redundancy analysis (RA) based on MBIST and provided control signals. Subsequently, rows and column pivots and restoration requests were serviced based on a precedence list that was arranged using the compared activities. Therefore, results were justified using the presented algorithm that was quite active as the repair rate was higher up to 4% assessed with the penalty of an area of the Essential Spare Pivoting, and few nominal area penalties were assessed with Essential Spare Pivoting.

## 3. Proposed Built-In Self-Test and Built-In Self-Repair Methodology for SRAM

The fault is inserted and tested to improve any system’s performance before it is marketed. The process of defects being added to the system is known as fault injection. In system-on-chip (SOC) design, memory occupies a larger space, and any memory defects affect the overall output of the system-on-chip. Thus, spare rows and columns are included in the memory. This manuscript proposes Deep Q-learning (DQL) and BSLFSR for built-in self-test and Built-In Self-Repair for SRAM. The proposed DQL-based MBIST is used to check the memory array unit, and faults are inserted into the memory using the DQL fault injection process. Subsequently, the fault memory is repaired after inserting faults in the memory cell using the BSLFSR-based Built-In Self-Repair model. The proposed block of built-in self-test and Built-In Self-Repair for Embedded Memories is shown in Figure 1.

The notations used in Built-in Self-test and Built-In Self-Repair for Embedded Memories block are summarized in Table 1.

The proposed architecture model involves various blocks such as a start register, test pattern generator, test controller, comparator, memory under Test (MUT), output response recorder, and BISR. The test pattern generator and test controller are performed based on the proposed DQL method. Moreover, faults are injected using the proposed DQL model, and the fault memory is repaired using the proposed BSLFSR.

### 3.1. Built-In Self-Test (BIST) Using Deep Q-Learning Algorithm

Typically, BIST is a practical integrated circuit with low cost incorporated with SRAM memories to test the fault occurring during the memory’s read or write operation. It eliminates the need for an expensive and time-consuming external hardware module known as Automated Test Equipment (ATE). Additionally, the architecture of BIST involves various structures than the external ATE because it is very expensive. In general, a built-in self-test circuit involves various blocks such as a buffer circuit that act as a level shifter, an amplifier circuit for amplifying the fault signal, the operational amplifier is utilized for boosting the weak signals and operates as 2nd stage amplifier, and a comparator circuit for evaluating the results of fault-free and faulty SRA Memory. Furthermore, the memory test controller works based on the Deep Q-learning mechanism to improve fault coverage and is utilized for detecting the coupling faults. The proposed Deep Q-learning mechanism generates the test patterns and acts as a test controller for injecting faults into the memory. After injection of defects in the memory cell, fault memory is fixed using BSLFSR for the BISR model.

Test pattern generator

It works on the DQL model that creates the patterns needed for injecting faults and propagating the effect to outputs. Let the initial state of DQL be as $K_{0}$, and the agent receives the state observation $\left(K_{s}\right)$ in a step (s) that takes action based on the policy $\pi \left(K_{s},A_{s}\right)$. The Q-function or a state-action value function from the state K and action A under the policy π is described in Equation (1),
(1)$$Q^{\pi }(K,A)=\sum_{t=0}^{\infty }\phi^{t}R_{s+t}|K_{s},A_{s}=A$$

The reward function is denoted as *R*, which states a scalar reward for a given state or action, ϕ representing the discount factor used to prefer the test patterns. Additionally, the model’s loss function during training is reduced using Equation (2),
(2)$$L(\varphi )=\left(y(\varphi^{-})-Q{\left(K,A,\varphi \right)}^{2}\right)$$
where the loss function is mentioned as L(φ) of DQN parameters φ and $y(\varphi^{-})$ represents the generated test patterns given in Equation (3),
(3)$$y(\varphi^{-})=\left(R+\phi \max Q(K^{\prime },A^{\prime },\varphi^{-}\right)$$

To track the test patterns $\varphi^{-}$ is used in the DQR mentioned in Equation (4),
(4)$$\varphi^{-}\leftarrow g\varphi +(1-g)\varphi^{-}$$
where g denotes the parameter for identifying test patterns that is mentioned as g<<1. Thus, to enhance the training stability, weights of the target network are periodically updated through repetitions. Hence, the generated test patterns are stored and applied during BIST execution. In this, the patterns are randomly generated by a DQL-based test pattern generator that act as test patterns. Moreover, an essential emphasis of register design is less area, which is created with as many various patterns as possible.

Test Controller

In this work, the memory test controller involves registers for starting the test controller to record the failed information. Hence, the test controller starts once the start signal is programmed in the start register. Additionally, the start register contains reset, resume, clock, start, stop, halt-on-error, and memory ID. Additionally, the test pattern generator works based on the DQL method that generates patterns applied in the block of MUT during the read operation. Moreover, the read data is compared with generated patterns during the testing operation. At last, the memory ID, faulty cells, defective cell counts, and failed address of the memory under test are recorded in the output response recorder. Moreover, the failed data are given to a built-in repair block for using spare memory in the place of faulty cells.

### 3.2. Memory under Test (MUT)

Additionally, the generated patterns are applied to MUT during the read and write operations. In this, various states are considered, such as idle state, write 0 at the memory location (w0), read 0 write 1 state (r0w1), read 1 write 0 read 0 (r1w0r0), write 0 read 0 write 1 (w0r0w1), read 1 word 0 (r1w0),read 0 (r0) and fail record status. Hence, the test controller in an idle state waits to start the signal if it is received once, then jumps to write 0 state (w0) and starts the memory test operation. Thus, MUT is full of zero patterns when it jumps to another state as r0w1. Here, read and write operations are performed using a test controller. Correspondingly, the test controller executes every operation successively at every state.

### 3.3. Comparator

In this model, the comparator compares the output and the pattern from the MUT block. Additionally, the read data are compared with preferred patterns while performing the testing under-read operation. If the comparison outcome fails, the test controller jumps to the failure-registration level, storing the comparison outcome and returning to another address location.

### 3.4. Output Response Recorder

It is responsible for verifying the output responses, which means the computer response applied to test vectors should be examined. Further, the decision is taken whether the system is faulty or not. Here, the faulty cells, fail address, memory ID, and MUT defective cell count records are stored in the output response recorder. Subsequently, the predefined patterns are given to memory under test, and the attained response is noted in the output response recorder.

Every operation is completed and shows the outcome in the status of fail record state. Additionally, Test Controller detects the failure memory ID, faults location, faulty cell, and faulty cell count. Subsequently, simulation results of the proposed model compute various fault types, such as Stuck Fault (SF), Coupling Faults (CF), Read Destructive Fault (RDF), Write Destructive Faults (WDF), Transition Coupling Faults (TCF), Static Coupling Faults (SCF), Disturb Cell Coupling Faults (DCCF), Incorrect Read Faults (IRF), Deceptive Read Destructive Faults (DRDF), Idempotent Coupling Faults (ICF).

### 3.5. Fault Modeling of SRAM

The description of possible approaches by which SRAM fails is defined as the fault modeling of SRAM. Next, the proposed DQL model is aimed to inject various categories of faults into the SRAM memory. Furthermore, possible defects in SRAM are explained as follows:

Stuck Fault (SF)

SF is defined as the single-cell fault where the logic value in the SRAM memory cell is stuck at 0/1.

Coupling Faults (CF)

CF is one of the categories of fault that occurs in SRAM cells due to its interaction with other cells. This originates under double cell faults, and two states are an increasing state that specifies zero to one, and the falling state sets 0 to 1.

Read Destructive Fault (RDF)

RDF belongs to a single-cell fault that is occurred in SRAM cell values into inverted, and then the resultant incorrect value is getting when a read operation is executed in the cell. If memory is zero, reads zero takes place, and cell memory becomes one; hence it is specified as 0r0/1/1 and if memory is one, read one takes place, and the cell memory becomes zero. Thus it is defined as 1r1/0/0.

Write Destructive Faults (WDF)

WDF is the category of single-cell faults that is non-transition kinds of operations, cells in the memory start to flip. Furthermore, in WDF, two various methods are presented among write’s destructive fault. If memory is zero, write zero occurs, and the cell becomes one. Hence it is specified as 0w0/1/ and if memory is one, write one takes place, and the cell becomes zero; accordingly, it is defined as 1w1/0/.

Transition Coupling Faults (TCF)

TCF is a double cell fault, and here transition does not occur if the write transition operation is used among cells of the victim word. The fault is 0W1/0/ for up transition, and the fault is 1W0/1 for down transition.

Static Coupling Faults (SCF)

SCF is a double cell fault that occurs in 0/1, and it is attained in the cells of the victim word because of forcing the aggressor word when 0 to 1 value is given to the cell.

Disturb Cell Coupling Faults (DCCF)

DCF is one of the double cell fault categories that occur when writing operation or read operation is achieved over the aggressor word, resulting in cell disturbance of the victim word.

Incorrect Read Faults (IRF)

IRF falls under the double cell fault category when a read operation occurs in the SRAM cell, gets the incorrect value, and the memory cell state remains stable. If memory is zero, read zero occurs, but when cell memory becomes zero, the reading process turns one and is specified as 0r0/0/1. If memory is one, read one takes place, but when cell memory becomes one, it returns to zero and is specified as 1r1/1/0. Finally, the read operation yields the aggressor value.

Deceptive read destructive faults (DRDF)

DRDF is a single-cell fault that occurs because the value in the cell is reversed and gets the accurate value while performing the read operation. If memory is zero, read zero takes place, and cell memory becomes one when the reading process turns into zero, which is specified as 0r0/1/1, and if memory is one, read one takes place. Cell memory becomes zero, then read turns into one specified as 1r1/0/0, i.e., the value is reversed after the read operation.

Idempotent Coupling Faults (ICF)

ICF is a double cell fault that happens when forced by the cell of aggressor word consists of higher (0–1)/lesser (0–1) transition of write operation for obtaining final value (0 or 1) in the cell is presented in the victim word.

Finally, the fault information (various faults) and memory failure information (faulty cell, faulty location, and memory ID) is transmitted to the BISR block.

### 3.6. Built-In Self-Repair (BISR) Utilizing Bit-Swapping-Based Linear Feedback Shift Register (BSLFSR)

This manuscript deals with two individual processes described in the flowchart in Figure 2. The first is a Bit-Swapping-based linear feedback shift register, and the second is the BSLFSR-BISR method to perform a Built-In Self-Repair process. The BSLFSR is a process of improving actual LFSR performance next spare rows and columns are used instead of faulty cells using the BSLFSR-BISR method. The BS-LFSR framework focuses primarily on minimizing power dissipation by reducing the conversion process in the conventional LFSR without compromising its efficiency and effectiveness. The proposed BSLFSR acts as the repair analyzer due to low power consumption. Initially, the memory failure information, such as faulty cell, faulty location, memory ID, and fault information, are transferred to the Built-In Self-Repair block to repair defective cells of the failed memory. BSLFSR-BISR involves spare memory or row-column block and redundancy logic (RL).

The redundancy management logic is utilized to store the faulty addresses found in the memory test process. Further, it compares the defected addresses with previously saved addresses in the fault table in case of multiple faults. When read and write operations of memory matches, the BSLFSR-BISR method starts working, and information is accessed via the spare memory. Simultaneously the repair analysis block access to the failure data and measures the repair signature stored in the register. Furthermore, it is likened to defected addresses and prior saved addresses in the fault table in case of multiple faults. Then, the address is stored only for the read and write memory operations that are not available in the fault table. Built-In Redundancy Analysis decides spare row or column allocation related to the information of faulty cell numbers in a specific address. Moreover, the Built-In Self-Repair block works in the proposed Bit-Swapping-based linear feedback shift register principle as mentioned in the flowchart.

In BSLFSR-BISR, the pre-charge technique is used for repair analysis or fault diagnosis. Here, the XOR gate is replaced, which lessens the delay and power in the circuit. Hence increasing operating frequency is attained along with the increasing performance of the circuit. Moreover, the standard repairing works in a simple rule. If a row contains more faults, it is said to be repaired, and if the column contains various faults, then various rows and columns are fixed. The faulty cell count is established using a memory test controller that is taken as a reference by BSLFSR for measuring the repair signature.

Moreover, a predefined threshold value uses spare rows and columns that are decided based on the counts of defected cells in defected rows or columns. Hence, the predefined threshold value equals ‘2’ or greater than ‘2’. If the row defect count is greater than ‘2’ or equal to ‘2’, then the spare row will allocate first; or else, a spare column will allocate. * The checking process will continue if the spare memory (row or column) allocation is over. If it is non-zero, the spare row allocates the checking process by continuing until it reaches the null state. Therefore, the spare memory is increased based on the faulty cell count. Hence this method significantly provides a memory test and fault repair solution by using the control flow mentioned in the flowchart.

## 4. Results and Discussion

This section describes the performance of the DQL-BSLFSR-FD method. Here, the simulation and synthesis outcomes are attained by utilizing Mentor Graphics and Xilinx ISE 14.5 Design Suite. Additionally, design and implement FPGA architecture by MBIST and Built-In Self-Repair hardware structure for Static Random Access Memory and tested using Verilog. The performance metrics, such as area, power, delay, slice register, maximum operating frequency, and clock period, are analyzed. The FPGA performance of the DQL-BSLFSR-FD method is likened to existing approaches. The proposed method’s FPGA performance is compared with the memory BIST with optimized BISR for fault detection (FPGA-MBIST-OBISR-FD) [22], SRAM-based Physically Unclonable Function (PUF) with Hot Carrier Injection (HCI) for fault detection (FPGA-PUF-HCI-FD) [23], and the Essential Spare Pivoting (ESP) based Local Repair Most (LRM) (FPGA-ESP-LRM-FD) [27], respectively.

### 4.1. Performance Metrics

The performance metrics, such as area, delay, power, slice register, maximum operating frequency, and clock period, are analyzed to validate the efficiency of the proposed method.

#### Calculation of Power Consumption

The average power consumption of the proposed DQL-BSLFSR-FD model is calculated using Equation (4)
(5)$$P_{avg}=\lambda V_{DD}^{2}(C_{L}.C_{f})$$
where the clock frequency of the model is represented as $C_{f}$, load capacitance is denoted by $C_{L}$, activation factor is represented as λ, and the supply voltage is denoted as $V_{DD}^{}$.

### 4.2. Simulation Outcomes

The proposed DQL-BSLFSR-FD method achieves better performance in terms of area, power, and delay when compared with other existing methods. In this work, simulation and synthesis outcomes are attained using Mentor Graphics and Xilinx ISE 14.5 Design Suite, where the design is implemented on Virtex-5 FPGA. During testing, various faults are injected into the memory using DQL with test pattern generation. The test patterns and fault injection into the memory are controlled through the test benches written to test the memory during testing. Various test cases are considered for testing the memory by applying different test patterns and injecting various faults for negative testing. There are 256 different test patterns are required for 8-bit size memory. Additionally, the generated patterns are used to detect every possible fault in the memory. The injected faults and the generated test patterns by the proposed DQL-BSLFSR-FD approach are obtained. Here, the numbers of defected cells are identified by the DQL-BSLFSR-FD approach to allocating the spare rows and spare columns.

Figure 3 shows the test pattern generation outcome by the proposed method. Various test patterns are generated to test the MUT for faults. The DQL-BSLFSR-FD model will test the memory and create the fault report. It also counts the number of faulty cells in each memory address and simultaneously updates the fault count register. The row- and column-wise details of defected count cell are computed, and the outcome of the faulty cells are stored in the fault count register by the proposed DQL-BSLFSR-FD method and are shown in Figure 4.

Once the faulty cell information is obtained from the test controller, the repair process begins. The simulation output for the repair of the defective cells in rows and columns is shown in the screens taken of the simulator after getting the output. The spare rows and columns are allocated for the faulty cell information received by the DQL-BSLFSR-FD block. Subsequently, the spare row allocation in SRAM memory using the DQL-BSLFSR-FD method is shown in Figure 5. Similarly, the spare column allocation in SRAM memory using the proposed DQL-BSLFSR-FD method is represented in Figure 6.

Table 2 shows the comparison of simulation outcomes for the proposed and existing approaches. Next, the slices of the proposed method are 24.6% and 32.7% lower than ELM-AE-FD and BIST-PPI-FD approaches, and the slice flip-flop count of the proposed method attains 18.9% and 26.7% lower than the ELM-AE-FD and BIST-PPI-FD approaches. Subsequently, the LUT of the proposed method is 15.7% and 26.7% lower than ELM-AE-FD and BIST-PPI-FD methods. The minimum clock period of the proposed approach is 11.85% and 18.7% lower than the existing ELM-AE-FD and BIST-PPI-FD methods. Furthermore, the maximum operating frequency of the proposed method is 23.5% and 29.5% increased than the existing ELM-AE-FD and BIST-PPI-FD approaches. Moreover, the total power consumption for the proposed method attains 34.9% and 26.7% lower than the existing ELM-AE-FD and BIST-PPI-FD techniques.

### 4.3. Comparative Analysis of Performance Metrics Using FPGA

The proposed structure is designed and tested by utilizing Verilog explanations that are targeted for Virtex-5, xc5vlx30 FPGA. The performance of FPGA implementation of the proposed DQL-BSLFSR-FD method is compared with existing approaches, such as memory BIST with optimized BISR for fault detection (OBISR-FD) [28], PUF-HCI-FD [29], and ESP-LRM-FD [30] FPGAs, respectively.

Figure 7 shows the comparison of the area for the proposed DQL-BSLFSR-FD method with the existing (OBISR-FD) [28], PUF-HCI-FD [29], and ESP-LRM-FD [30] FPGAs approaches. The area calculation of the proposed FPGA-DQL-BSLFSR-FD method provides 85%, 72%, and 79% lower area than the existing approaches, such as MBIST-OBISR-FD, PUF-HCI-FD, and ESP-LRM-FD FPGAs, respectively.

Figure 8 compares the slice register of the proposed FPGA-DQL-BSLFSR-FD method with the existing MBIST-OBISR-FD, PUF-HCI-FD, and ESP-LRM-FD FPGA approaches. The slice register of the proposed DQL-BSLFSR-FD FPGA method provides an 85%, 80%, and 50% lower slice register than the existing approaches, such as MBIST-OBISR-FD, PUF-HCI-FD, and ESP-LRM-FD, FPGA, respectively.

Figure 9 shows the maximum operating frequency comparison for the DQL-BSLFSR-FD method with existing MBIST-OBISR-FD, PUF-HCI-FD, and ESP-LRM-FD FPGA approaches. The maximum operating frequency of the proposed DQL-BSLFSR-FD method provides 67.8%, 33.33%, and 56.7% higher maximum operating frequency than the existing approaches, such as MBIST-OBISR-FD, PUF-HCI-FD, and ESP-LRM-FD FPGAs, respectively.

Figure 10 compares the minimum clock period of the DQL-BSLFSR-FD method with the existing MBIST-OBISR-FD, PUF-HCI-FD, and ESP-LRM-FD FPGA approaches. The minimum clock period of the proposed DQL-BSLFSR-FD method provides a 40%, 25%, and 50% lower minimum clock period than the existing approaches, such as MBIST-OBISR-FD, PUF-HCI-FD, and ESP-LRM-FD FPGAs, respectively.

Figure 11 shows the comparison of power consumption for the proposed DQL-BSLFSR-FD method with the existing MBIST-OBISR-FD, PUF-HCI-FD, and ESP-LRM-FD FPGA approaches. The power consumption of the proposed DQL-BSLFSR-FD FPGA method provides 91.66%, 83.33%, and 75% lower power consumption than the existing method, such as MBIST-OBISR-FD, PUF-HCI-FD, and ESP-LRM-FD FPGA, respectively.

Figure 12 shows the delay comparison for the DQL-BSLFSR-FD approach with existing MBIST-OBISR-FD, PUF-HCI-FD, and ESP-LRM-FD FPGA approaches. Then, the delay of the proposed DQL-BSLFSR-FD method provides 91.53%, 86.84%, and 80.77% lower delay than the existing approaches, such as MBIST-OBISR-FD, PUF-HCI-FD, and ESP-LRM-FD FPGA, respectively.

Figure 13 compares the access time of the proposed DQL-BSLFSR-FD method with existing MBIST-OBISR-FD, PUF-HCI-FD, and ESP-LRM-FD FPGA approaches. The access time of the proposed DQL-BSLFSR-FD method provides 34.5%, 42.6%, and 26.7% lower access times than existing approaches, such as MBIST-OBISR-FD, PUF-HCI-FD, and ESP-LRM-FD FPGAs, respectively.

As per today’s demand of consumers to use a large volume of the memory in their devices, this research study provides the solution to test the memories embedded in the SoC and attempt to improve the product yield. This study offers the solution for the regress testing SRAM in SOC-based development. It proposed the memory test method through the proposed MBIST controller to find more and possibly all kinds of defects. Further, it provides a powerful solution to repair the faculty memory cell or faulty memory location supplied by the test block. It uses a spare memory cell to replace the detected defective cells to repair the memory. The design is targeted to the FPGA platform, and the obtained results are targeted to the power consumption, speed, and area overhead. The obtained results are reasonable if we compare them with the existing studies targeting the same parameters. It is discussed in the literature and the above result section and observed from the results obtained on the FPGA that the proposed method is better in terms of delay, power, and area parameters. The comparison of results of the existing studies [28,29,30] is carried out, the obtained results are tabulated, and the performance graphs are plotted as shown in the charts of the result section. The contribution to the memory test and repair is significant, and the existing work’s performance is acceptable compared to the studies considered in this research study.

## 5. Conclusions

This manuscript proposes the Deep Q-learning (DQL) with Bit-Swapping-based linear feedback shift register (BSLFSR) for Fault Detection for SRAM. Furthermore, the proposed DQL-based MBIST effectively generates test patterns and injects faults into the memory. Here, the BSLFSR-based BISR model is utilized to repair the faults’ injection in memory. Therefore, the BSLFSR model performs the redundancy analysis to detect the faulty cells and use the spare rows and columns. Finally, the proposed DQL-BSLFSR-FD model has attained 85%, 72%, 79% lower area, 85%, 80%, 50% lower utilization of slice register, 91.53%, 86.84%, 80.77% lower delay, 91.66%, 83.33%, 75% lower power consumption, 67.8%, 33.33%, 56.7% higher maximum operating frequency, and 40%, 25%, and 50% lower minimum clock period than the existing approaches, such as MBIST-OBISR-FD, PUF-HCI-FD, and ESP-LRM-FD FPGAs, respectively.

## Figures and Tables

**Figure 1 micromachines-13-00971-f001:**
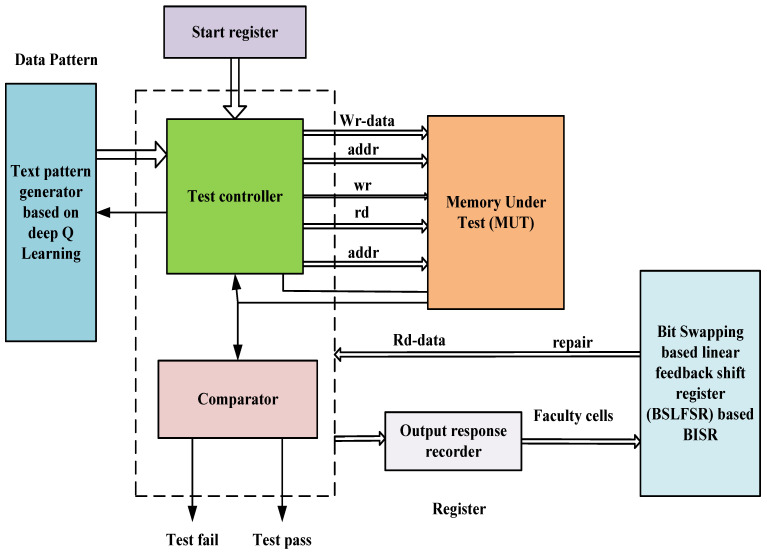
Block diagram of built-in self-test and Built-In Self-Repair for Embedded Memories.

**Figure 2 micromachines-13-00971-f002:**
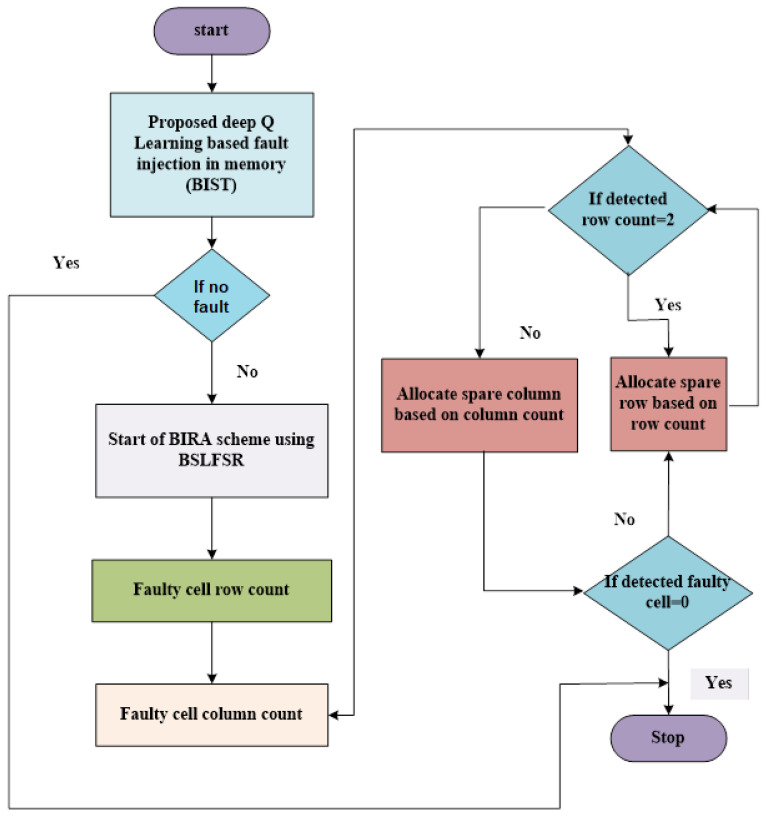
Flow chart of proposed method.

**Figure 3 micromachines-13-00971-f003:**
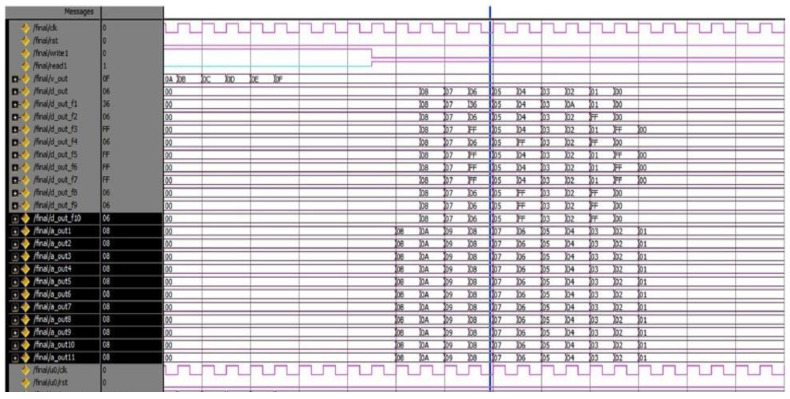
Test pattern generation using the DQL-BSLFSR-FD model.

**Figure 4 micromachines-13-00971-f004:**
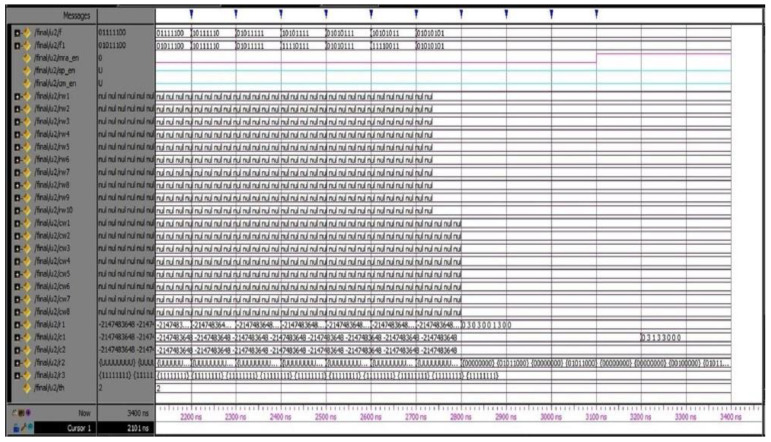
Defected count cell details in row and column using the DQL-BSLFSR-FD model.

**Figure 5 micromachines-13-00971-f005:**
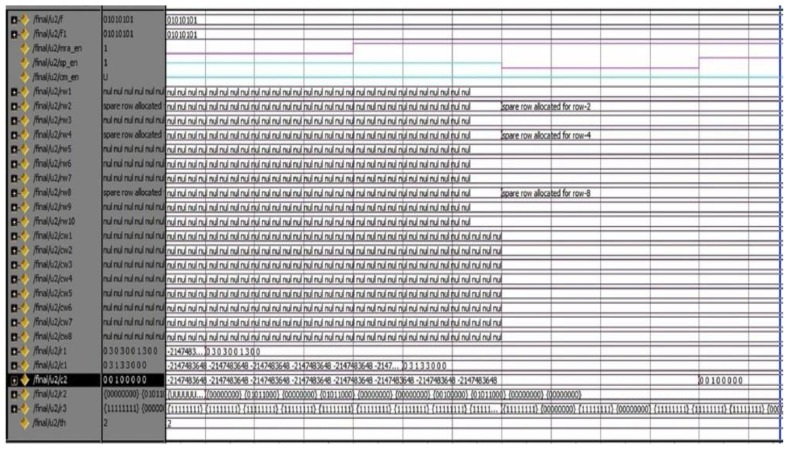
Allocation of spare rows using the DQL-BSLFSR-FD model.

**Figure 6 micromachines-13-00971-f006:**
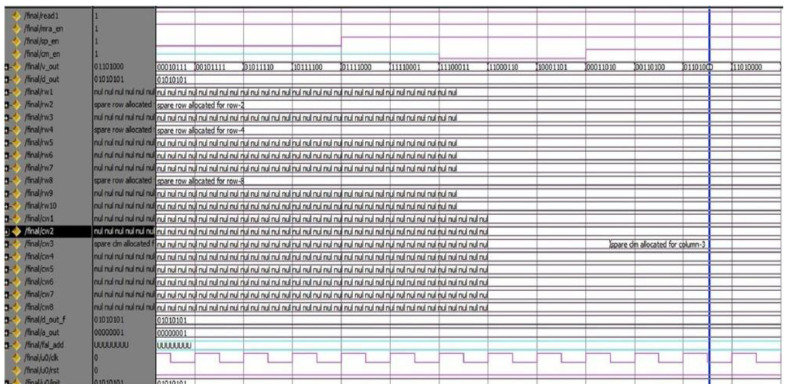
Allocation of a spare column using the DQL-BSLFSR-FD model.

**Figure 7 micromachines-13-00971-f007:**
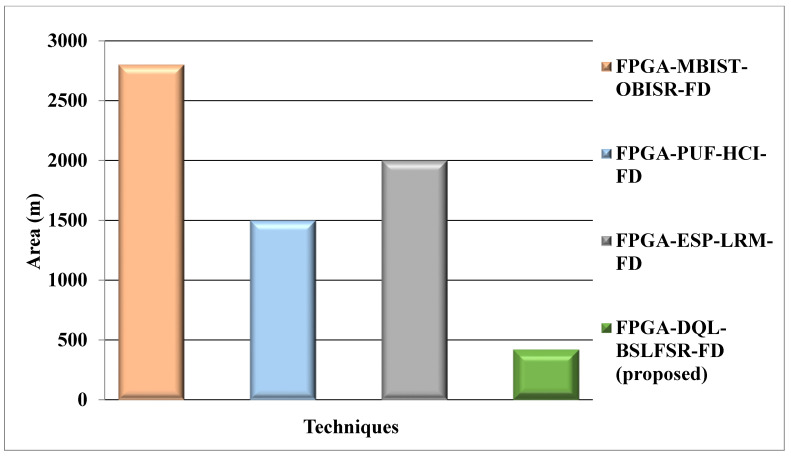
Area comparison for the proposed and other existing approaches.

**Figure 8 micromachines-13-00971-f008:**
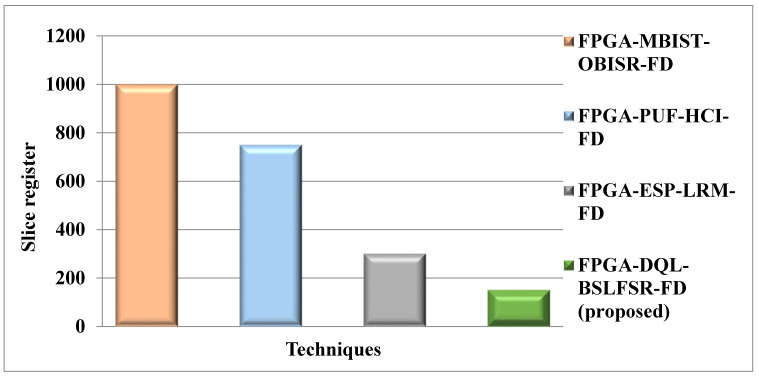
Comparison of slice register utilization for the proposed and other existing methods.

**Figure 9 micromachines-13-00971-f009:**
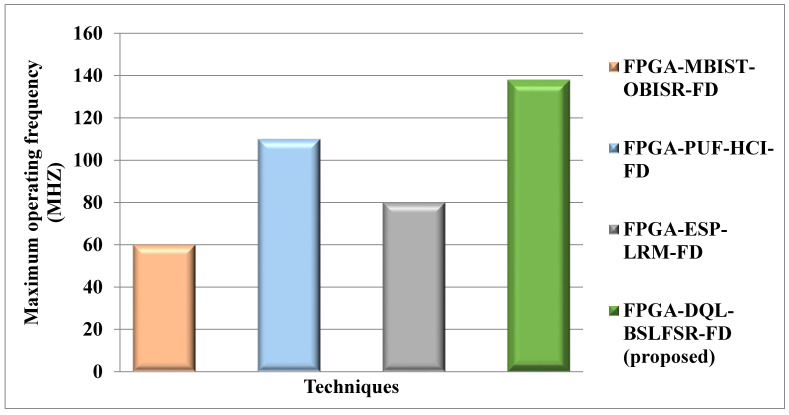
Comparison of maximum operating frequency of the presented and other methods on FPGA.

**Figure 10 micromachines-13-00971-f010:**
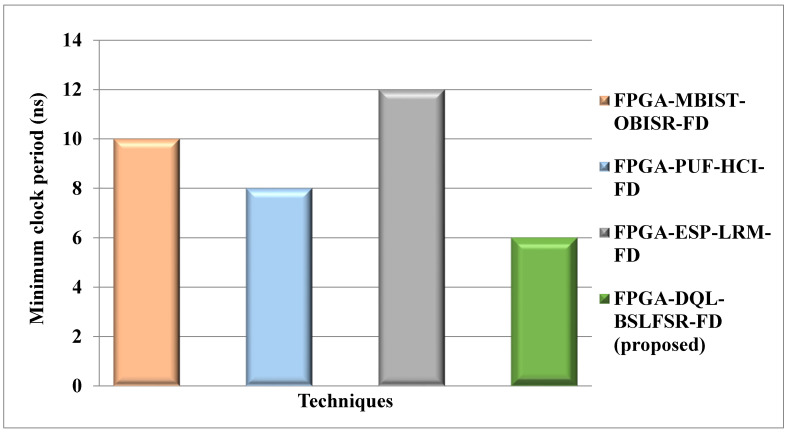
Comparison of a minimum clock period.

**Figure 11 micromachines-13-00971-f011:**
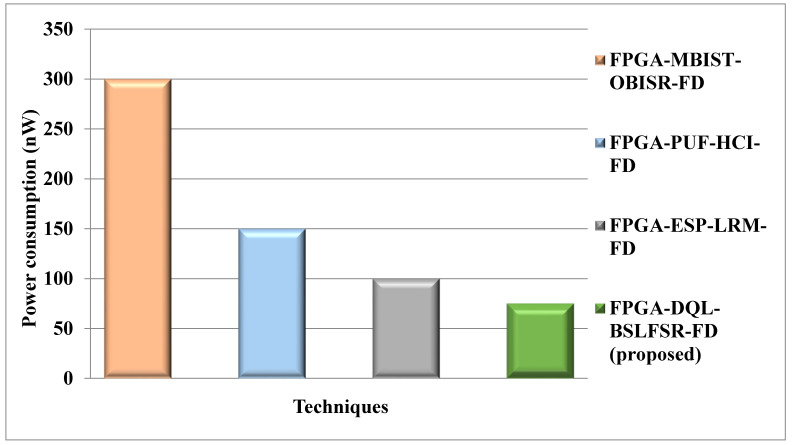
Power consumption comparison for the proposed and other existing methods.

**Figure 12 micromachines-13-00971-f012:**
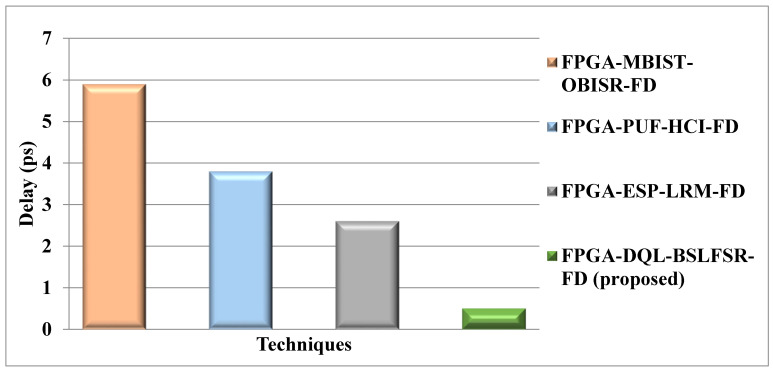
Comparison of delay for proposed and other approaches.

**Figure 13 micromachines-13-00971-f013:**
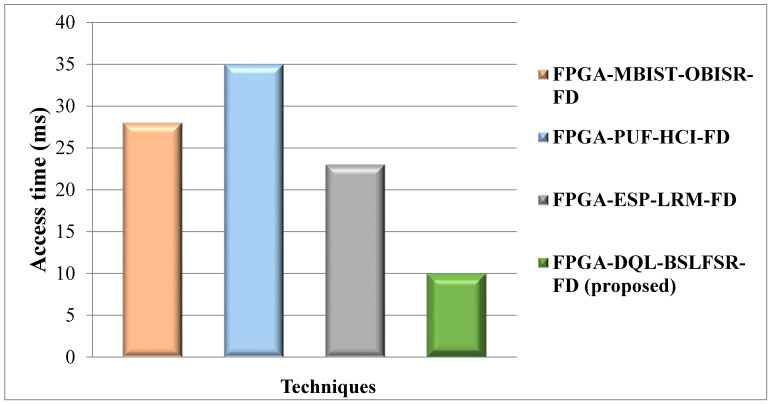
Comparison of access time between the proposed and other methods.

**Table 1 micromachines-13-00971-t001:** Description of the notation used in the Test and Repair process.

Operations	Description
Wr-data	Write the data to the memory location whose address is given.
addr	Indicates the address of the memory location where the data of the memory is going to access.
wr	Write enable signal to write into the memory.
rd	Read enable signal indicating the read operation from memory.
Rd-data	The read data bus contains the read data from the given memory location.
w0	Write the logic value ‘0’ to the memory location.
w1	Write the logic value ‘1’ to the memory location.
r0	Read the logic value ‘0’ from the memory cell.
r1	Read the logic value ‘1’ from the memory cell.

**Table 2 micromachines-13-00971-t002:** Simulation outcome comparison.

Performance Metrics	ELM-AE-FD [26]	BIST-PPI-FD [27]	DQL-BSLFSR-FD (Proposed)
Slice numbers	48	54	37
Slice flip-flop numbers	51	68	46
Look Up Table (LUT) numbers	91	87	70
Minimum clock period (ns)	9.5	8.9	6.5
Maximum frequency (MHz)	114	124.6	140.6
Total power consumption (nW)	156.6	134.7	74.7

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
