# Peer review of "Deep Q-Learning with Bit-Swapping-Based Linear Feedback Shift Register fostered Built-In Self-Test and Built-In Self-Repair for SRAM"

_micromachines, 2022, doi:10.3390/mi13060971_

Round 1

Reviewer 1 Report

Dear Authors,

Your manuscript is interesting however there are certain areas for improvement. One of the main issues is English usage, which is not of sufficient quality in several passages and makes the paper hard to read. Another big issue is related to the contribution to the state of the art. In the present form, it is unclear why your DQL-BSLFSR method performs better than the other solutions and the improvement seems marginal. Please try highlight more clearly the advantages and limits of your proposed methods with respect to the literature.

Please see my additional detailed comments.

Abstract

“Including redundancy is popular and widely used in a fault-tolerant method for memories.” -> please mention the origin of faults you aim at correcting

today's memories of large sizes -> today's large-size memories

11.85%, 18.7% lower minimum clock period, 23.5%, 29.5% maximum operating frequency -> these describe the same metric twice, please choose either frequency or period.

1.     Introduction

memory systems design in the system on chip -> memories in system on chips

It contains a lesser feature size -> It contains a smaller feature size

Peugeot -> Please clarify. For example, you could mention the chip name and it is main function.

utilizing Mentor Graphics and Xilinx ISE 14.5 design suite. -> This is an obsolete CAD suite, why did you choose such an old tool? Where you motivated by the specific FPGA (Virtex-5) you used for implementation?

2.     Related work

Please arrange the related work chronologically and provide a more quantitative discussion. For example, purely quantitative sentences like

“Therefore, the designed model achieved lower power consumption and higher reliability but the operating

time of the model was high.”

Should be avoided in favour or quantitative statements citing relevant metrics.

3.     Proposed Built-in Self-test and Built-in Self-Repair Methodology for SRAM

“Automated Test Equipment because” -> you could use the ATE acronym, which you have just defined

SF is defined as the single-cell -> Please clarify.

Redundancy logic block -> redundancy management logic (?)

4.     Results and discussion

better performance in performance metrics like area -> better performance in terms of area

I suggest to drop Fig. 3,4,5 and 6 as they do not seem to convey much information and they are very difficult to read. In case you decide to retain them please try to improve their comment in the text.

(in Table 1) Minimum clock period (s) -> I suppose the unit is nanoseconds (ns)?

Table 1  tabulates -> Table 1 shows

Please try to provide a more meaningful caption to figure 7.

The whole subsection 4.3 is very hard to read. There are several bar plots and little or no commentary. Perhaps it would be helpful to spend a few lines describing the differences of the implementations you are comparing. Also, please clean up the naming, all implementations include the FPGA prefix, therefore it can be dropped. Moreover, you could try to find an easier and more intuitive naming.

Author Response

Many thanks for your valuable suggestions to improve the quality of our manuscript. We tried our best to improve the readability of the text and corrected the English language of the manuscript. The suggested changes are incorporated into the text of the manuscript. We hope now the manuscript is as per your expectation.

Reviewer 2 Report

The authors focus their study on introducing a deep Q-learning with bit swapping based linear feedback shift register for fault detection supporting the static random access memory. Towards checking the memory array unit, the authors used the deep Q learning based memory built-in self test and also several types of faults are inserted into the memory using the deep Q-learning fault injection process. The manuscript is overall well written and easy to follow and the authors have well thought out their main contributions. The provided theoretical analysis is concrete, complete, and correct and the authors have provided all the intermediate steps in order to enable the average reader to easily follow it. The provided numerical results are also rich in order to show the pure operation and the performance of the proposed framework. The authors are encouraged to consider the following suggestions provided by the reviewer in order to improve the scientific depth of their manuscript, as well as they need to address the following comments in order to improve the quality of presentation of their manuscript. Initially, the provided related work in sections one and two needs to be substantially revised in order to be presented by using more summative language and identify the research gap that the authors try to address. Furthermore, in Section 2, the authors need to discuss several existing approaches that have been introduced in the literature and exploit the physical unclonable functions, such as Artificially Intelligent Electronic Money, doi: 10.1109/MCE.2020.3024512, in order to deal with the fault detection problem. In Section 3, the authors need to include a table summarizing the main notation that has been used in the paper. In the flow chart of the proposed method, the authors need to discuss the flow of information and control. In section 4, the authors need to provide some additional numerical results of comparative nature in order to quantify the drawbacks and benefits of the proposed approach. Furthermore, in Section 4, the authors need to discuss the implementation cost of the proposed framework. Finally, the overall manuscript needs to be checked for typos, syntax, and grammar errors in order to improve the quality of its presentation.

Author Response

Many thanks for your valuable comments and appreciating words on our manuscript. We tried our best to respond to your comments to improve the quality of our paper. The required changes are incorporated in the text of the manuscript. We hope now the manuscript is as per the standard of the journal.

Round 2

Reviewer 2 Report

The authors have addressed in detail the reviewers comments.